# Different Changes in Adipokines, Lipid Profile, and TNF-Alpha Levels between 10 and 20 Whole Body Cryostimulation Sessions in Individuals with I and II Degrees of Obesity

**DOI:** 10.3390/biomedicines10020269

**Published:** 2022-01-26

**Authors:** Wanda Pilch, Anna Piotrowska, Joanna Wyrostek, Olga Czerwińska-Ledwig, Ewa Ziemann, Jędrzej Antosiewicz, Mariusz Zasada, Małgorzata Kulesa-Mrowiecka, Małgorzata Żychowska

**Affiliations:** 1Faculty of Physiotherapy, Institute for Basic Sciences, University of Physical Education, 31-571 Krakow, Poland; wfpilch@op.pl (W.P.); anna.piotrowska@awf.krakow.pl (A.P.); olga.malgorzata.czerwinska@gmail.com (O.C.-L.); 2Institute of Physical Culture, State University of Applied Sciences, 33-300 Nowy Sacz, Poland; joanna.wyrostek1@gmail.com; 3Department of Athletics, Strength, and Conditioning, Poznan University of Physical Education, 61-871 Poznan, Poland; ziemann@awf.poznan.pl; 4Department of Bioenergetics and Exercise Physiology, Medical University of Gdansk, 80-210 Gdansk, Poland; jant@gumed.edu.pl; 5Department of Biological Foundations of Physical Culture, Kazimierz Wielki University, 85-091 Bydgoszcz, Poland; mariusz_zasada@vp.pl; 6Department of Rehabilitation in Internal Diseases, Faculty of Health Sciences, Jagiellonian University Medical College, 31-008 Krakow, Poland; m.kulesa-mrowiecka@uj.edu.pl

**Keywords:** obesity, chronic inflammation, leptin resistance, whole-body cryostimulation

## Abstract

Obesity is associated with chronic inflammation. While cold therapy influences the pro/antioxidative status of an individual, by affecting adipokine levels and the lipid profile, the effect of body mass index (BMI) on the response to cold exposure is unclear. We analyzed the link between BMI and the differences in effects of whole-body stimulation, depending on the number of treatments, on specific physiological parameters in men. Twenty-seven non-active men were divided into three groups: N (*n* = 9, BMI < 24.9), IOb (*n* = 9, BMI 30.0–34.9), and IIOb (BMI ≥ 35.0). The subjects participated in 20 3-min cryochamber sessions (−120 °C), 1/day, 5 days/week. Body composition was analyzed before and after treatment. Blood adiponectin (ADP), leptin (LEP), and tumor necrosis factor alpha (TNF-alpha) levels, and the lipid profile were analyzed three times: at baseline and up to 2 h after 10 and 20 sessions. The 20 treatments caused significant changes in body composition. Between 10 and 20 whole-body cryostimulation (WBC) sessions, a significant decreased was observed in the LEP and TNF-alpha levels. No significant changes in the lipid profile were noted. However, a positive tendency to regain the metabolic balance in adipose tissue was apparent in the IOb group in the tested period (decreased TG levels, increased HDL levels or the HDL/LDL ratio, and significantly decreased visceral adiposity index levels). Collectively, for people with obesity increasing the number of treatments above the standard 10 should be recommended.

## 1. Introduction

In highly developed countries, the level of physical activity in a population is limited. This, among other factors, impacts the severity of diseases of civilizations, such as obesity. The number of individuals with a body mass index (BMI) over 25 is continually increasing, and today it is approximately four times higher than in the 1980s. Obesity is a chronic inflammatory disease, strongly associated with other diseases of civilizations, especially type 2 diabetes [1], certain types of cancer and cardiovascular diseases, and depression [2,3]. It is characterized by low plasma levels of adiponectin (ADP). Low ADP levels are, in turn, associated with autoimmune diseases [4] and the development of hypertension [5]. Imbalanced diet [6], low physical activity [7], and genetic predisposition [8] are among the most common causes of obesity.

Maintenance of an appropriate adipose tissue mass is important for human health because of its participation in many processes, including thermogenesis, heat isolation, energy storage, and, especially, endocrine function [9]. ADP and leptin (LEP) are the best known adipokines produced by adipose tissue. In obese individuals, the expression of ADP receptors (AdipoR1 and AdipoR2) is reduced, and excessive serum LEP levels (hyperleptinemia) or LEP resistance have been reported [10]. Since LEP participates in food intake, and regulates metabolism and the degree of adiposity on the brain level, its circulating levels must be precisely regulated; however, LEP levels are often altered in obese individuals [9]. Further, while LEP has to cross the blood–brain barrier to exert its effect on the brain, blood–brain barrier alterations have been reported in obese individuals [10]. Finally, obese individuals can suffer from hypoadiponectinemia accompanied by elevated LEP levels [11,12,13]. Animal studies have provided a lot of interesting data on regulation of leptin secretion. Leptin could downregulate its own expression by an indirect, non-autocrine mechanism [14]. Furthermore, the role of the sympathetic system in inhibiting leptin expression and secretion by beta-3 adrenergic receptors was well documented [15,16,17].

Based on the available literature, a reduction in adipose tissue mass increases serum adipokine levels [18], ameliorates insulin sensitivity, and reduces inflammation [19]. ADP has pro-apoptotic and anti-proliferative properties, which is especially important in the context of individuals with tumors. Obesity-related adipose tissue dysfunction worsens the prognosis of cancer by increasing the risk of metastasis [20,21]. Modulation of the serum adipokine and LEP levels, resulting in insulin and LEP sensitivity, provide hope for weight reduction and prevention of concomitant disease in obese individuals. However, in extremely obese individuals, the implementation of physical activity, the main approach to reducing adipose tissue mass, may be problematic, and any treatment that aids mass reduction and influences the adipokine and lipid profiles is highly recommended.

According to numerous recent reports, whole-body cryostimulation (WBC) has many beneficial effects on human health, including on body composition [22], blood circulation in tissue [23], antioxidant status [24,25], hematological markers [26], glucose homeostasis [27], and others. Nagashima et al. [28] reported that cryogenic treatment increases the metabolic rate and heat production by stimulating the activity of the sympathetic nervous system or non-shivering thermogenesis. This impacts adipose tissue metabolism by mobilizing intracellular triglycerides (TG) and enhancing oxidation or glycogenolysis [29,30] in white and brown fat tissue. There are many indications of the effect of WBC on the lipid profile; however, data regarding the adipokines ADP and LEP are limited and concern different numbers of sessions, WBC procedures, or BMI of participants. The effect of WBC on ADP and LEP levels was only demonstrated in two studies. Ziemann et al. [13] reported no changes in ADP and LEP levels in obese subject (BMI > 30) after 10 WBC sessions, and Lubkowska et al. [31] reported similar observations for overweight subjects following 6 months of moderate aerobic activity combined with 20 WBC sessions. Based on the available data, in physically active men, 10 WBC sessions lead to beneficial changes in TG levels and depend on a decrease in the levels of total cholesterol (TC) and low-density lipoprotein (LDL), and an increase in the levels of high-density lipoprotein (HDL) [13]. A decrease in TC and LDL levels following 30 min of exercise with WBC applied twice a day for 5 consecutive days was reported in another study [32]. Similar findings were reported by Rymaszewska et al. [33] for TC, TG, and LDL levels. The authors postulated that lower BMI may be linked to a greater improvement of the lipid profile by WBC. The results achieved by the cited authors do not answer the question of whether the amount of WBC should be considered depending on BMI and whether increasing treatments in people with increased adipose tissue will give positive results, noted for people with normal BMI.

The anti-inflammatory effect of WBC is well documented [19,34,35,36]. Nonetheless, to the best of our knowledge, a comparison of the effects of different numbers of WBC sessions on the lipid profile and adipokine levels in individuals with class I and II obesity had not yet been investigated. Further, the published studies employ different WBC treatment methodologies, and involve participants with different levels of physical fitness or obesity. Furthermore, while the amount of kcal consumed during an experiment can influence the experimental findings, dietary analyses of study participants are scarce. We aimed to address these points in the current study, specifically focusing on the effect of BMI on the beneficial effects of WBC in humans. We hypothesized that effects of 20 WBC on body composition will be positive and BMI-dependent. Furthermore, it is possible that for positive changes in adipokines or lipid profiles more than 10 cryo-sessions will be effective and BMI-dependent. Our findings have implications for obese individuals with LEP resistance.

## 2. Materials and Methods

### 2.1. Study Protocol Overview

Volunteers from the Association of bariatric patients at the Bariatric Clinic of Collegium Medicum UJ in Krakow, who expressed willingness to participate in the study, were subjected to medical assessment as per the inclusion and exclusion criteria (vide infra). Following anthropometric measurements, blood was collected by a qualified person for biochemical analyses. All participants received dietary instructions from a dietician, considering their body mass and nutritional energy demands. After a medical examination and the consent of the doctor to patriciate in the experiment, a series of sessions in a cryochamber began. Before each session, the blood pressure was measured. Individuals with values exceeding 160/100 mmHg were excluded from further participation in the study. Immediately after the 10th and 20th WBC session, blood was re-drawn for analysis. Then, 24 h after 20 WBC sessions, body composition and circumferences were reassessed.

### 2.2. Ethics

The project was approved by the Bioethics Committee at the Regional Medical Chamber in Krakow (Poland) (no. 122/KBL/OIL/2017) and registered in Australian New Zealand Clinical Trials Registry (ANZCTR) as a clinical trial (no. ACTRN12619000524190, registered on 2 April 2019).

### 2.3. Study Group

Forty volunteers qualified for the study but only 27 completed the study protocol. The participants were divided into three groups: those with class II obesity (*n* = 9, BMI > 34.9, IIOb group), those with class I obesity (*n* = 9, BMI between 30.0 and 34.9, IOb group), and those with a healthy body weight (*n* = 9, BMI up to 24.9, N group). Characteristics of the study groups are presented in Table 1.

The following inclusion criteria were used: BMI within the accepted range, and no contraindications to cryostimulation procedures, as confirmed by the qualifying physician. The exclusion criteria were the use of anti-inflammatory drugs, statins, or fibrates, and diagnosed diabetes.

The participants were informed about the purpose and methods of the study, and about the possibility of withdrawing at any stage, provided written consent to participate in the study, and had access to their data. Further, they were instructed not to change the form and frequency of any recreational physical activity throughout the duration of the study.

### 2.4. Anthropometric Measurements

Anthropometric measurements were taken before and after the WBC sessions by an experienced athropometrist in accordance with ISAK guidelines [37]. Body mass (BM) was determined using an IOI-353 body composition analyzer (Yawon, Korea). Anthropometric tape was used to measure the waist (WC) and hip (HC) circumferences. Body height (BH) was only measured prior to the commencement of the WBC sessions, using a height measuring device (Seca 216, Hamburg, Germany) with an accuracy of 5 mm. BH was then used to calculate the BMI. The body fat percentage (PBF), body fat mass (FM), and lean body mass (LBM) were determined using a BodygramPLUS body composition analyzer (Akern BIA 101, Pontassieve, Florence, Italy). In addition, body adiposity index (BAI) and visceral adiposity index (VAI), anthropometric indicators used for body fat assessment, were calculated, using the following formulas [26,27]:$$\mathrm{VAI}=\frac{\mathrm{WC}\left[\mathrm{cm}\right]}{39.68+\left(1.88\times \mathrm{BMI}\right)}\times \left(\frac{\mathrm{TG}}{1.03}\right)\left(\frac{1.31}{\mathrm{HDL}}\right)$$
$$\mathrm{BAI}=\frac{\mathrm{HC}\left[\mathrm{cm}\right]}{\mathrm{BH}\left[m\right]\times \sqrt{\mathrm{BH}\left[m\right]}}-18$$

VAI, viscelar adiposty index; WC, waist circumference; BMI, body mass index; TG, triglicerides; HDL, high density lipoproteins; BAI, body adiposity index; HC, hip circumference; BH, body height.

### 2.5. Dietary Analysis

Before the start of the study, a dietician from the Bariatric Clinic Collegium Medicum UJ set a balanced diet for all subjects, considering the BM, level of physical activity, and BMI (Table 2). To verify compliance, the participants were asked to keep a detailed 5-d diet diary using an Album of Photos of Products and Dishes (Food and Nutrition Institute, Warszawa, Poland) [38]. Based on the obtained data, the calorific value and individual dietary components were calculated using the Diet 6.0 program (Food and Nutrition Institute, Warszawa, Poland).

### 2.6. WBC Procedure

The study participants underwent a series of 20 cryostimulation sessions at −120 °C at Malopolska Cryotherapy Center Krakow (Poland). Each session lasted 3 min, and each participant underwent 1 session a day, 5 day a week. Each session was conducted with the participation of a medical doctor supervising the study. The WBC treatments were preceded by a 30 s adaptation period in an atrium at −60 °C. The stay in the cryochamber was monitored by an appropriately trained employee of the facility and contact with the study group was maintained via a camera and a voice system. Before entering the cryochamber, the subjects were informed about the recommended moving and breathing style during WBC.

### 2.7. Blood Collection and Biochemical Analysis

To assess the initial condition of tested parameters in participants, blood samples were collected up to 1 h before the first treatment. To compare the difference between 10 and 20 treatments, blood samples were collected twice: up to 1 h after 10 and 20 treatments. Samples containing 6 mL of venous blood were collected from antecubital fossa veins by a laboratory diagnostician. For each participant, blood collection was performed at the same time of day. Since cryotherapy cannot be done on an empty stomach, blood sampling was performed at least 6 h after the last light meal. In accordance with the applicable standards, a laboratory diagnostician collected venous blood with elbow flexion into test tubes (Vacumed^®^ system, F.L. Medical, Torreglia, Italy). Tubes containing the blood and a clot activator were centrifuged (3000× *g*, 10 min, 5 °C) within 15–20 min of blood collection. The obtained supernatant was pipetted, transferred to 1.5 mL microtubes, and placed in a low-temperature freezer (−60 °C) until further analysis. ADP, LEP, and tumor necrosis factor alpha (TNF-alpha) levels were determined in blood serum without traces of hemolysis using enzyme-linked immunosorbent assay (ELISA) and an ELISA microplate reader (Chromate 4300, Awareness Technology Inc., Palm City, FL, USA). Assay sensitivity was 0.7 ng∙mL^−1^ for LEP, 0.27 ng∙mL^−1^ for ADP, and 0.7 pg∙mL^−1^ for TNF-alpha. Intra and inter assay [CV%] variability was as follows: for LEP: 3.14% and 7.5%; for ADP: 5.95% and 8.7%; for TNF-alpha: 5.2% and 7.4%.

Lipidograms were obtained using an enzymatic–colorimetric method (Cobas C501, Roche Diagnostics, Basel, Switzerland) using glycerolphosphate oxidase for TG determinations; cholesterol esterase and oxidase for TC; magnesium (II) ions, dextran sulfate, and cholesterol esterase and oxidase for HDL cholesterol (HDL-C); and a homogeneous method with cholesterol esterase and oxidase in the presence of surfactants for LDL cholesterol (LDL-C). Non-HDL cholesterol (nonHDL-C) fraction was calculated using the formula:nonHDL-C [mmol·L^−1^] = TC [mmol·L^−1^] − HDL-C [mmol·L^−1^]

### 2.8. Statistical Analysis

The data were analyzed using Excel 365 (MS Office, Redmond, WA, USA) and GraphPad Prism 6.0 ((www.graphpad.com, accessed on 19 January 2022) GPM6-277417-RLMI-DE56F). The results are presented as an arithmetic mean with standard deviation. The type of distribution was verified using the Shapiro–Wilk test and it revealed that all variables were normally distributed. Paired t-test and one-way ANOVA and post Tukey test were used to determine differences between treatments in the various groups. Differences between groups at all time points were analyzed using an unpaired t-test and a two-way analysis of variance (ANOVA) with repeated measures. For all tests, the significance level was set at *p* < 0.05. For statistically significant results, Cohen’s d was calculated and interpreted as follows: 0–0.2, small effect; 0.5, medium effect; and above 0.8, large effect.

In order to calculate sample size, we used data from a pre-study, in which levels of indicators were measured before and after 10 and 20 cryotherapy treatments in both groups (*n* = 5 per group). The sample sizes for IOB, IIOB, and N groups (*n* = 9) were determined based on the mean and SD within a group to provide 80% power and *p* = 0.05 (Es = 1.33; σ = 1.15).

## 3. Results

### 3.1. General Characteristics of Participants

General characteristics of the participants at baseline and 24 h after 20 WBC sessions are presented in Table 1.

After 20 WBC sessions, a significant decrease compared to rest values in most parameters was observed in all groups, except for BMI and HC in the N group. Cohen’s d analysis of the baseline values indicated pronounced differences between the groups. Two-way ANOVA confirmed differences in the response of the groups to 20 WBC sessions (significant differences for time, row data, interaction, and subject). Further, the analysis of the parameter differences before vs. after WBC indicated that individuals with class I obesity were the most sensitive to 20 WBC sessions.

#### 3.1.1. Differences in Tested Parameters and Indexes at Baseline

All results obtained from blood analysis were not corrected for changes in plasma volume, which were very small and statistically insignificant.

LEP levels were above the health-related reference interval in the IOb and IIOb groups at baseline/rest values and increased with an increasing BMI. In the N group, the LEP levels were within the reference range, similarly to ADP levels in the N and IOb groups but were below the reference range in the IIOb group. However, there were no significant differences in the ADP levels between the three groups. TNF-alpha levels did not significantly differ between the groups at baseline and all values were within the reference range (i.e., below 12.4 pg/mL) (Figure 1A).

TC, TG, non-HDL, and LDL levels in the IIOb group were significantly higher than those in the N group. However, all these parameters showed the tendency to increase with an increasing BMI. The TC levels in the IIOb group were above the reference range (3–5 mmol/L). At baseline, the TG values were within the reference range only in the N group. There were no differences in HDL between groups (Figure 1B).

HDL and LDL levels in the IIOb group were outside the reference range (below 1.20 mmol/L and above 2.59 mmol/L, respectively). The mean HDL levels in the IOb group were close to those in the N group and were within the reference range (≥1.2 mmol/L). In the IIOb group at baseline, the HDL levels were significantly lower and the LDL levels were significantly higher than those in the N and IOb groups. The HDL/LDL ratio in the IIOb group was significantly lower than that in the other groups (Figure 1B).

Finally, BAI levels in the IOb and IIOb groups were significantly higher than those in the N group (Figure 1C). Baseline VAI levels in the IOb and IIOb groups were higher than those in the N group. HDL/LDL ratio was significantly lower in IIOb compared to the N group.

#### 3.1.2. Comparison of Changes in LEP, ADP, and TNF-Alpha after 10 and 20 WBC in Three Groups Differing in BMI

No differences in the effect of 10 and 20 WBC on the N group LEP levels were apparent, while a significant decrease in LEP levels was observed between 10 and 20 WBC sessions in the IOb and IIOb groups (Figure 2A). Further, between 10 and 20 WBC sessions, no effect was observed in ADP levels in individual groups (Figure 2B). The additional 10 sessions of WBC caused in significant decreased the TNF-alpha levels in IOb and IIOb.

#### 3.1.3. Comparison of Changes in TC, TG, and Non-HDL-C after 10 and 20 WBC in Three Groups Differing in BMI

A significant reduction in TC levels was observed between 10 and 20 WBC sessions in the IOB and IIOb group (Figure 3A). In addition, additional 10 WBC sessions reduced the TG levels in the IOb group, bringing them to within the reference range (under 1.7 mmol/L). The TG levels in the IIOb group after 10 and 20 WBC sessions were significantly higher than those in the N group. No differences caused by number of WBC sessions were observed for the TG levels in the N and IIOb groups (Figure 3C).

The tendency to decrease in nonHDL-C levels was noted between 10 and 20 WBC, especially in IOb and N groups. The nonHDL-C levels in the IIOb group after 20 WBC sessions were higher than those in the N and IOb groups (Figure 3C).

#### 3.1.4. Comparison of Changes in HDL, LDL, and HDL/LDL Ratio after 10 and 20 WBC in Three Groups Differing in BMI

Between 10 and 20 WBC there were no significant differences in the HDL levels, LDL levels, and the HDL/LDL ratio in any group; however, these values tended to increase between 10 and 20 WBC sessions in the IIOb and IOb groups (Figure 4A–C). The mean HDL levels in the IOb group were close to those in the N group and were within the reference range. In the IIOb group after 20 WBC sessions, the HDL levels were lower, and the LDL levels were significantly higher than those in the N group. The HDL/LDL ratio in the IIOb group was significantly lower after 10 and 20 sessions than that in the N group (after 20 WBC compared to N and IOb).

#### 3.1.5. Changes in BAI and VAI after 20 WBC Sessions

Finally, BAI levels in the IOb and IIOb groups were significantly higher than those in the N group and were unaffected by 20 WBC sessions (Figure 5a). Baseline VAI levels in the IOb and IIOb groups were higher than those in the N group; however, they were significantly decreased in the IOb group after 20 WBC sessions (Figure 5b) (*p*
< 0.05).

The directions of changes in adipokines and lipids in tested groups after 10 and 20 WBC sessions are summarized in Table 3.

## 4. Discussion

### 4.1. Changes in the Adipokines and TNF-Alpha Profile during the Experiment

Our results confirm a disturbed lipid metabolism in people in the IOb and IIOb groups, relying on raised levels of most parameters (except of HDL). As presented herein, rest levels of this adipokine increase with an increasing degree of obesity, with significant differences between the three groups after 10 WBC sessions. Following 10 sessions caused a further decrease in LEP, especially in the IIOb group. After 20 sessions, no significant differences between IOb, N, and IIOb groups were noted. Thus, the main finding of the current study is the significant decrease in LEP levels in the IOb and IIOB groups between 10 and 20 WBC sessions. In the N group, with a recommended LEP health related reference interval, no effect at the same time was observed. Many authors have reported increased LEP levels in obese subjects [19,39,40,41]. Nonetheless, studies regarding the effect of cold treatment on LEP levels are limited or involve different cold exposure protocols [31,32]. For instance, Puerta et al. [42] tested the effects of acute cold exposure (18 h at 6 °C) on the expression of *LEP* and *ADP* in brown and white adipose tissues in rat. They found a significant decrease in *LEP* and *ADP* levels in brown adipose tissue, and a significant decrease in *LEP* levels but no changes in *ADP* levels in white adipose tissue. The authors postulated that *LEP* gene expression is inhibited by cold conditions and does not impact *ADP* gene expression in white and brown adipose tissue [42]. Further, Imbeault et al. [43] reported a decrease in LEP levels after 2 h of cold exposition in humans. It is well known that obese subjects exhibit LEP resistance [10,44]. Hence, any treatment that results in a decrease in LEP levels or causes LEP sensitivity might be a good strategy for preventing diseases of civilizations, often associated with obesity [45,46]. In the current study, the largest decrease in LEP levels was recorded in the IIOb group after 20 WBC. The drop of LEP levels in obese individuals (the IOb and IIOb groups) in response to following 10 WBC treatment might be beneficial considering its effect on muscle shivering and glucose uptake and metabolism [47,48]. Moreover, it is possible that WBC could positively influence impaired beta-3 adrenergic activity in obesity [16].

In the current study, between 10 and 20 sessions no changes were observed in ADP levels. According to the literature, a similar effect in response to 10 WBC was reported by Ziemann et al. [13] and Lubkowska et al. [24], who combined cold exposure (20 WBC sessions) with a 6-month exercise program in overweight men. According to many reports, ADP levels in obese individuals, especially ones with visceral obesity, are lower than those in non-obese subjects [47,49,50,51]. These differences were noted in the current study; however, a large individual variability in adiponectin concentration was apparent in the IIOb group.

Many authors postulated the anti-inflammatory properties of WBC, as demonstrated by a reduction in TNF-alpha levels in response to WBC. Our findings showed a significant decrease in TNF-alpha between 10 and 20 WBC sessions in IOb and IIOb groups. Interestingly, the changes in TNF-alpha levels were not accompanied by an increase in ADP levels. Thus, the decline in TNF-alpha levels progressed with the number of WBC sessions. According to the available literature, TNF-alpha suppresses ADP gene expression in adipocytes [52]. Therefore, the anti-inflammatory response to WBC is of major importance to obese individuals, however, for these people following 10 sessions produced good results. Further, high TNF-alpha levels are associated with insulin resistance [19]. The data presented herein are in agreement with reports on the reduction in levels of other inflammatory indicators, such as C-reactive protein [22] and interleukin 1β [13], in response to WBC.

### 4.2. Changes in Body Composition Induced by 20 WBC Sessions

In this study we also evaluated the effect of WBC on body composition (differences at baseline and 48 h after the last WBC session). During the experiment, no significant changes were observed in, BM, PBF, HC, and WC in all groups compared to rest values, except for BMI and HC in the N group. The largest changes were observed in the IOb group. A decrease in subcutaneous fatty tissue thickness, probably associated with its metabolic activity and a reduction in adipocyte volume under cold stress conditions [33], was likely the main reason for the observed changes in body composition. These observations are in agreement with those of Pilch et al. [22], but contrast with those of Ziemann et al. [13] and Lubkowska et al. [53]. These authors reported a reduction in the BM, FM, and PBF after 20 WBC in obese men [22], and no changes in body composition after 10 WBC in individuals with a BMI under 30 [13]. Furthermore, Lubkowska et al. [31] reported no changes in body composition after an exercise program incorporating 20 WBC sessions in overweight subjects. Differences in the number of WBC treatments, a mixed WBC-exercise regime, baseline physical fitness, or the diet could all explain these contrasting findings. The participants of the current study were not physically active and observed a balanced diet, which was not reported on in the other studies. Further, while the individuals in the IIOb group (mean BMI 42.7) showed large disturbances in the lipid profile and adipokine secretion, 20 WBC sessions did not significantly affect body composition. According to the available literature, subcutaneous fat might act as a barrier to heat loss, influence thermoregulatory ability, and act as a thermal insulation layer [54]. Accordingly, obese individuals might lose heat more slowly than non-obese individuals, and an increase in their metabolic rate under cold conditions might be less pronounced than that in non-obese individuals. This may explain the observed differences in the effect of WBC on IOb and IIOb groups.

### 4.3. Changes in the Lipid Profile between 10 and 20 WBC Sessions

Obesity is associated with a decrease in HDL levels [55], and an increase in LDL and TG levels [31], as confirmed in the current study. Of note, the HDL and LDL levels in individuals in the IOb group were close to those in the N group. Generally, the TC and TG levels tended to decrease in response to following 10 WBC, but only in the N and IOb groups (significant changes for TC). Further, the HDL levels tended to increase slightly in these groups upon following WBC with a simultaneous lowering in LDL, consequently the HDL/LDL ratio tended to increase in the groups. Lubkowska et al. [53] reported significant changes in TG levels after 10 and 20 WBC sessions in physically active healthy men, with a significant decrease in TC and LDL levels, and a simultaneous increase in HDL levels, after 20 WBC sessions. Herein, in our study similar tendencies were apparent in the N and IOb groups between 10 and 20 WBC sessions. However, no effects of following 10 WBC stimulus were observed for participants in the IIOb group, and no positive changes in the lipid profile were observed in these individuals even after 20 WBC sessions. Ziemann et al. [13] observed positive changes in the lipid profile already after 10 WBC sessions in physically active men. The physical fitness level and BMI reference norm are important for the effectiveness of WBC, as has been also noted by several other authors [56,57].

All changes in adipokine (especially leptin) and TNF-alpha levels as well as in lipid profile may be associated with a response of the sympathetic nervous system. It is known that catecholamines, which inhibit production and secretion of TNF-alpha and LEP, are also modulated by the sympathetic nervous system. Moreover, animal studies showed that mutations in beta-3 adrenergic receptors lead to lowered metabolic rate in obese subjects [58].

BAI and VAI did not change significantly in response during an additional 10 sessions WBC in participants of the current study; however, a significant decrease in VAI, indicating a reduced risk of cardiovascular disease, was apparent in the IOb group. In addition, the mean HDL and LDL levels, and the mean HDL/LDL ratio in the IOb group were close to the ones in the N group in the tested period. Interestingly, the largest changes in the adipokine levels and lipid profile in the IOb group were consistent with the largest changes in body composition in that group. It is possible that 20 WBC sessions are sufficient for individuals with healthy body mass and class I obesity with no major metabolic disturbances at baseline, however, positive changes in some parameters also occur in the IIOb group. In individuals with class II obesity, additional WBC sessions affect body composition, proinflammatory TNF-alpha and LEP levels, and HDL/LDL ratio. Our data shows a difference in response between 10 and 20 WBCs, however, the effects of greater numbers of treatments, which may be required for people with class II obesity, were not investigated in this study.

## 5. Conclusions

Our findings indicate that between 10 and 20 WBC sessions there are many positive changes associated with adipokines, lipid profile, or body composition. The drop in LEP levels could have resulted from a decreased LEP resistance, a desired effect, especially among obese subjects. The IOb group was the most sensitive to WBC, in terms of the fact that only BAI and ADP were not affected by following 10 WBC. Overall, 20 WBC sessions had a positive effect in individuals with minor disturbances in baseline values; however, further research is needed, especially involving obese individuals undergoing a higher number of sessions than that used in the current study.

## 6. Study Limitations

Our study has some limitations. We present only differences between subjects with different BMI during tested period. We did not investigate changes in tested indicators in people who did not use WBC at the same time. Moreover, the size of the studied groups is an additional limitation.

## Figures and Tables

**Figure 1 biomedicines-10-00269-f001:**
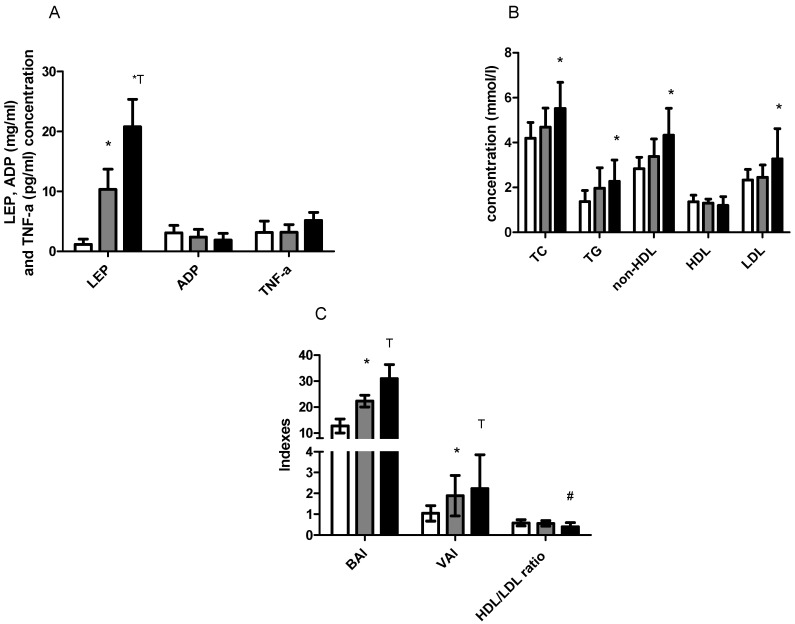
Blood concentration of LEP, ADP, TNF-alpha (**A**); TC, TG, non-HDL, HDL, and LDL (**B**); BAI, VAI, and HDL/LDL index (**C**) between N (white bars), IOb (gray bars), and IIOb (dark bars) at baseline. *—significant differences compared to the N group (*t*-test, ANOVA one-way). T—significant differences between N and IOb (*t*-test, ANOVA one-way. #—significant differences between N and IIOb (*t*-test, ANOVA one-way).

**Figure 2 biomedicines-10-00269-f002:**
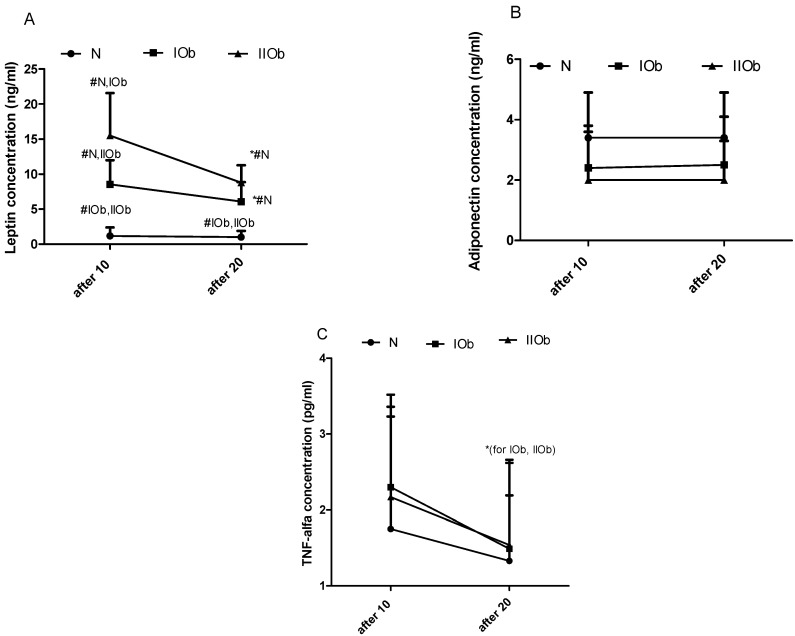
Changes in leptin (LEP) (**A**), adiponectin (ADP) (**B**), and tumor necrosis factor-alpha (TNF-alpha) (**C**) levels between the three groups after 10 WBC sessions and after 20 WBC sessions. Health related reference interval: LEP, 2.05–5.63 ng/mL; ADP, 2.0–13.9 ng/mL; TNF-alpha, no more than 12.4 pg/mL. Two-way ANOVA confirmed significant differences between the experimental groups (for time, row factor, and subject). *—significant differences between 10 and 20 WBC (*t*-test and ANOVA 2-way; *p* < 0.05). #—significant differences between groups in the same timepoints (*t*-test and ANOVA 2-way; *p* < 0.05).

**Figure 3 biomedicines-10-00269-f003:**
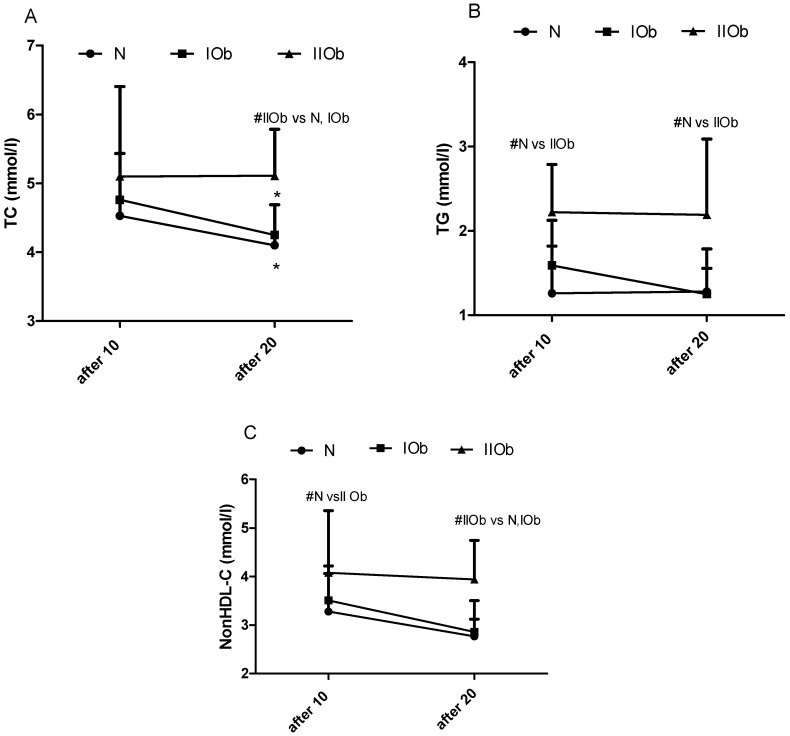
Changes in total cholesterol (TC) (**A**), triglycerides (TG) (**B**), and non-HDL-C (**C**) levels between the three groups after 10 and 20 WBC sessions. Normal ranges: TC, 3.0–5.0 mmol/L; TG, no more than 1.70 mmol/L; non-HDL-C, 3.8 mmol/L for low cardiovascular risk (<1% SCORE scale), <3.4 mmol/L for moderate cardiovascular risk (≥1% to <5% SCORE scale), <2.6 mmol/L for high cardiovascular risk (≥5% to <10% SCORE scale), and <2.2 mmol/L for very high cardiovascular risk (≥10% SCORE scale); *—significant differences in values between sessions within groups (*p* < 0.05); #—significant differences in values between groups in the same timepoints (*p* < 0.05). Two-way ANOVA revealed significant differences between the three groups for interaction, row factor, and time.

**Figure 4 biomedicines-10-00269-f004:**
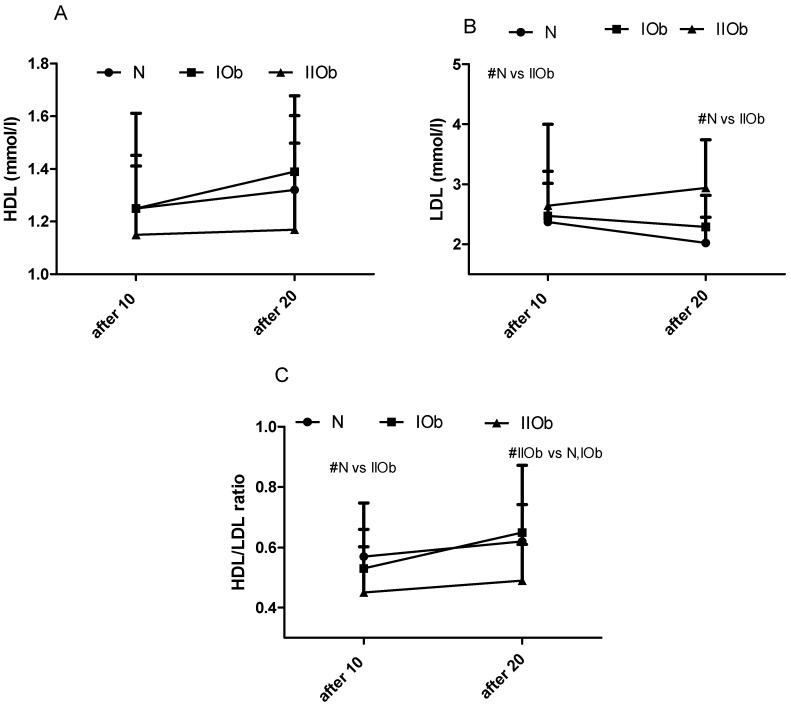
Changes in HDL (**A**), LDL (**B**), and HDL/LDL ratio (**C**) between the three groups after 10 and 20 WBC sessions. Normal ranges: HDL, more than 1.20 mmol/L; LDL, <3.0 mmol/L for low cardiovascular risk (<1% SCORE scale), <2.6 mmol/L for moderate cardiovascular risk (>1% to <5% SCORE scale), <1.8 mmol/L for high cardiovascular risk (>5% to <10% SCORE scale), and <1.4 mmol/L for very high cardiovascular risk (>10% SCORE scale); HDL/LDL ratio, >0.4 optimal, 0.4–0.3 moderate, <0.3 high. # significant differences between groups (*p* < 0.05).

**Figure 5 biomedicines-10-00269-f005:**
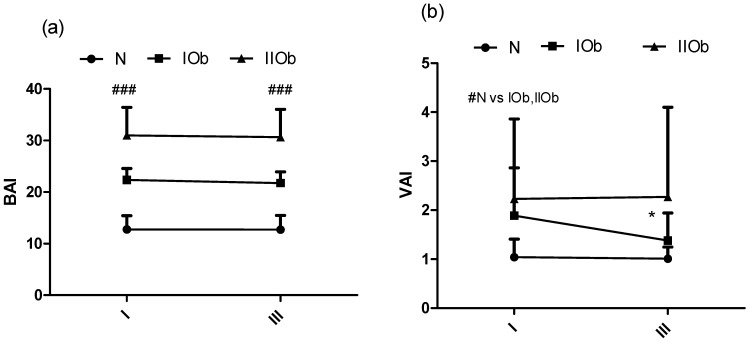
Changes in BAI (**a**) and VAI (**b**) between the three groups at rest (I) and after 20 WBC (III); health related reference interval BAI, <8% underweight, 8–21% healthy, 21–26% overweight, and >26 obese (normal ranges for men aged: 20–39 years); VAI, ≤2.52 for individuals less than 30 years old and ≤2.23 for individuals aged between 30 and 42 years. # significant differences between groups (*p* < 0.05). ### significant differences between all tested groups (*p* < 0.05). * significant decrease between I and II.

**Table 1 biomedicines-10-00269-t001:** The effects of 20 sessions of whole-body cryostimulation on body composition of obese and normal body weight male subjects.

	N (*n* = 9) (BMI < 24.9) Mean ± SD		IOb (*n* = 9) (BMI 30–34.9) Mean ± SD		IIOb (*n* = 9) (BMI > 35) Mean ± SD		Cohen’s d			ANOVA
							N vs. IOb	N vs. IIOb	IOb vs. IIOb	
Age (y)	21.4 ± 1.2 *		31.1 ± 3.8		28.4 ± 3.6					
BH (cm)	183.0 ± 5.3		179.7 ± 4.1		180.7 ± 2.0					
	Before	After	Before	After	Before	After				
BMI (kg/m^2^)	23.0 ± 1.5	22.5 ± 1.4	31.3 ± 1.5	30.0 ± 1.8 *	42.7 ± 8.0	42.1 ± 1.8 *	5.5	3.4	2.0	#
Δ	−0.4 ± 0.6		−1.2 ± 0.9		−0.5 ± 0.7					
BM (kg)	76.9 ± 6.2	76.1 ± 6.7 *	101.2 ± 9.0	97.3 ± 9.7 *	139.2 ± 24.8	123.7 ± 16.5 *	3.1	3.5	2.0	#
Δ	−0.8 ± 0.6		−4.0 ± 2.8		−1.5 ± 2.1					
HC (cm)	75.7 ± 4.8	75.6 ± 4.9	96.8 ± 7.4	95.3 ± 6.8 *	117.0 ± 12.5	116.0 ± 12.4 *	3.4	4.4	2.0	#
Δ	−0.1 ± 1.2		−1.4 ± 1.0		−0.8 ± 0.8					
WC (cm)	73.9 ± 2.4	73.4 ± 2.5 *	99.7 ± 1.4	97.1 ± 2.0 *	126.7 ± 17.0	123.7 ± 16.5 *	13.1	4.4	2.2	#
Δ	−0.4 ± 0.32		−2.6 ± 1.2		−2.2 ± 1.6					
PBF (%)	11.0 ± 2.4	9.8 ± 2.7 *	28.2 ± 2.0	26.2 ± 1.9 *	36.6 ± 6.9	34.9 ± 7.3	7.9	5.0	1.7	#
Δ	−1.2 ± 1.2		−2.0 ± 2.7		−1.3 ± 2.5					

BH, body height; BMI, body mass index; BM, body mass; HC, hip circumference; WC, waist circumference; PBF, percentage of body fat; Δ, change; *, significant difference before vs. after; #, significant difference for time, row data, subject, interaction.

**Table 2 biomedicines-10-00269-t002:** Analysis of the study participant diet, with the calorific value and percentage of energy obtained from protein, carbohydrate, and fat.

	N (*n* = 9) (BMI < 24.9) Mean ± SD	IOb (*n* = 9) (BMI 30.0–34.9) Mean ± SD	IIOb (*n* = 9) (BMI > 35.0) Mean ± SD
Energy (kcal)	2852.2 ± 130.1	3011.9 ± 431.1	3350.4 ± 91.1
Protein (% of energy)	24.3 ± 2.1	23.6 ± 1.8	24.1 ± 1.1
Carbohydrate (% of energy)	51.2 ± 11.3	53.5 ± 12.1	49.80 ± 10.1
Fat (% of energy)	24.5 ± 9.0	22.9 ± 1.8	26.1 ± 0.8

**Table 3 biomedicines-10-00269-t003:** Directionality of changes observed between 10 and 20 WBC sessions in the different groups (N, normal body composition; IOb, class I obesity; IIOb, class II obesity).

	Direction (between 10 and 20 WBC)		
	N	IOb	IIOb
Leptin	→	↓	↓
Adiponectin	→	→	→
TNF-alpha	↓	↓	↓
TC	↓	↓	→
TG	↓	↓	→
nonHDL	↓	↓	→
HDL	↑	↑	↑
LDL	↓	↓	→
HDL/LDL	↑	↑	↑
BAI	→	→	→
VAI	→	↓	→

↓, decrease; →, no change; ↑ increase.

## Data Availability

All data is held by the corresponding author and will be available upon request.

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
