# Peer review of "Different Changes in Adipokines, Lipid Profile, and TNF-Alpha Levels between 10 and 20 Whole Body Cryostimulation Sessions in Individuals with I and II Degrees of Obesity"

_biomedicines, 2022, doi:10.3390/biomedicines10020269_

Round 1

Reviewer 1 Report

The manuscript by Pilch et al. is interesting and provides new data regarding the role of BMI in the metabolic adaptation to cold.

Nevertheless, there is an important point that should be taken into consideration:

It is well known since 1996 (J Biol Chem 1996 Mar 8;271(10):5301-4 doi: 10.1074/jbc.271.10.5301; Diabetes. 1996 Dec;45(12):1744-9. doi: 10.2337/diab.45.12.1744) the role of the sympathetic system inhibiting leptin expression and secretion. This effect was found to be mediated by beta-3 adrenergic receptors in two other seminal articles: Endocrinology. 1996 Sep;137(9):4054-7. doi: 10.1210/endo.137.9.8756584; Life Sci. 1997;61(1):59-64. doi: 10.1016/s0024-3205(97)00358-5.

The effect of catecholamines inhibiting TNF-alpha production is known even at an earlier time: J Immunol. 1992 Jun 1;148(11):3441-5.

Moreover, impaired beta-3 adrenergic activity in obesity is also well known since 1994 (Mol Endocrinol. 1994 Apr;8(4):518-27. doi: 10.1210/mend.8.4.7914350). In fact, a mutation of the beta-3 adrenergic receptor was found to be associated with obesity and also the earlier appearance of type 2 diabetes (N Engl J Med. 1995 Aug 10;333(6):343-7. doi: 10.1056/NEJM199508103330603).

Therefore, regardless the effects of cryotherapy on lipid profile, since the effects of cryotherapy may be mediated by the sympathetic system, the work may be improved by studying sympathetic activity and the expression of beta-3 adrenergic receptors in white adipose tissue, because these data may explain the mechanism of the differences in the response to cryotherapy according to BMI. If this is not possible, at least these considerations should be taken into account in the introduction and discussion sections.

Author Response

Response to Rev. 1

We would like to many thanks for the reviewer for valuable comments that will help improve our manuscript.

The Reviewer wrote: “The manuscript by Pilch et al. is interesting and provides new data regarding the role of BMI in the metabolic adaptation to cold.”

Response`: Thank you very much for this comment.

The Reviewer wrote: “Nevertheless, there is an important point that should be taken into consideration”:

Response: Thank you very much for paying attention to the important issue concerning the influence of the sympathetic nervous system on LEP secretion and lipid metabolism. In the current version we have added in the introduction section:

Animal study has produced a lot of interesting data on regulation of leptin secretion. Leptin could be down-regulated its own expression by an indirect, non-autocrine mechanism (Slieker et al. 1996). Furthermore, the role of the sympathetic system in inhibiting leptin expression and secretion by beta-3 adrenergic receptors was well documented (Kossaki et al. 1996; Gettys et al. 1996; Breslof et al. 1997)”,

and in Discussion section:

Moreover, it is possible that WBC could positively influence on impaired beta-3 adrenergic activity in obesity (Collins et al.1994), but this sentence requires further research.” and „ All changes in adipokines (especially leptin downregulation), lipid profile or TNF-alfa may be associated with the changes in activity of the sympathetic system. It is known that catecholamines inhibiting TNF-alpha production, LEP production and secretion is also modulated by sympathetic nervous system and that obesity could be associated with the mutation in beta-3 adrenergic receptors, which could cause in lowered metabolic rate in obese subject (Waltson et al. , 1995).

Once again, thank you very much for all comments and your time dedicated to us.

Reviewer 2 Report

Abstract : indicate what TNF, WBC and VAI are standing for

Methods

It is not clear when were the blood specimens collected and this is the main problem of this manuscript. Were they at baseline at rest or not? Is there information and data missing?  It is difficult to understand -if I understood correctly - why the authors have only collected specimens after the 20th session and the 10th  session and not before.

I would suggest the authors to collect specimens before and after the first, the 10th and the 20th sessions and even organize the blood specimen collection for each participant at the time of the day. In their experimental setting the authors can only compare the data after the 10th and the 20th sessions and cannot explain any effect of the cryo with comparing the data at rest before any cold exposure and the data obtained after the 20th session of cryo.

It would also be of interest to check whether the observed changes are linked or not with hemoconcentration/hemodilution. Therefore, hematrocrite or the concentration of Hb would be of interest.

Were the blood collection organised at the time of the day. A chronobiological effect could have an influence on the authors data.

L165: at XXX ??

The authors should present for each analyte determination the CV intra and inter assays

Concerning the discussion, I would refrain the authors to discuss their blood analysis data if they are not collected at baseline. If this is the case, then the discussion is OK though a bit too long

Author Response

Response to Rev. 2

We would like to many thanks for the reviewer for valuable comments that will help improve our manuscript.

In details:

The Reviewer wrote:

“Abstract: indicate what TNF, WBC and VAI are standing for”

Response: Abbreviations have been explained as requested.

Methods

The reviewer wrote: “It is not clear when were the blood specimens collected and this is the main problem of this manuscript. Were they at baseline at rest or not? Is there information and data missing?  It is difficult to understand -if I understood correctly - why the authors have only collected specimens after the 20th session and the 10th session and not before.”

Response: Thank you very much for these comments. In present version we explained in more detail:

“Blood samples were collected in three timepoints: up to 1 hour before first treatment, up to 1 hour after 10 and after 20 treatments. Samples containing 6 ml of venous blood were collected from antecubital fossa veins by a laboratory diagnostician. For each participant blood collection was performed at the same time of day.”

The Reviewer wrote: “I would suggest the authors to collect specimens before and after the first, the 10th and the 20thsessions and even organize the blood specimen collection for each participant at the time of the day. In their experimental setting the authors can only compare the data after the 10th and the 20th sessions and cannot explain any effect of the cryo with comparing the data at rest before any cold exposure and the data obtained after the 20th session of cryo.”

Response: Thank you for this suggestion. We collected blood samples 3 time at the same time of day for each participant. Now it is corrected in method section.

The Reviewer wrote: “It would also be of interest to check whether the observed changes are linked or not with hemoconcentration/hemodilution. Therefore, hematrocrite or the concentration of Hb would be of interest.”

Response: Thank you very much for this comment. We have the results of Hct and Hb, and we calculated changes in plasma volume, however no significant changes were determinate. The scattering of the results was from 0.99 to 1.06. I can provide these results for request, but they not influence on obtained results for another parameters. We added sentence: “All results obtained from blood analysis were not corrected for changes in plasma volume, which were very small and statistically insignificant”.

The Revewer wrote: “Were the blood collection organised at the time of the day. A chronobiological effect could have an influence on the authors data”

Response: Thank you very much for this comment. All blood collections in each participant were performed at the same time of the day, afternoon.

The Reviewer wrote: “L165: at XXX ??”

We apologize for this mistake. The name was added in place of XXX.

The Reviewer wrote: “The authors should present for each analyte determination the CV intra and inter assays”

Response: Thank you very much for these comments. Requested data has been added to the manuscript.

The Reviewer wrote: “Concerning the discussion, I would refrain the authors to discuss their blood analysis data if they are not collected at baseline. If this is the case, then the discussion is OK though a bit too long”.

Response: Thank you very much for this comment. In main text we changed baseline to rest value in many points of manuscript.

Once again, thank you very much for all comments and your time dedicated to us.

Reviewer 3 Report

A generally well written manuscript. Some points to consider.

Line 43. A minor point but "ADP" usually refers to adenosine diphosphate. I wonder if an alternative abbreviation should be used to avoid any possible confusion. Similarly "WBC" often used for white blood cells.

Line 108"Nutritional energy demands".

Line 120. The subject numbers are small. Was a power calculation conducted? If so what sample size was required? 13 out of 40 is a substantial drop out rate. Did the drop outs have any characteristics in common that differed from the remainers? Is this a confounder. Please comment and justify.

Section 2.3 and Table 1. Although mentioned in the Abstract please state here that the participants were all male.

Section 2.4. Please provide a reference to the standardized method for circumferential measurements. Were measurements obtained by an ISAK accredited anthropometrist? Weight was measured with the IOI-353 BIA analyser. This provides body composition data yet another analyser, Akern 101, was used to obtain these data - why? Please explain. BIA is a predictive method and is notoriously population specific. Are the predictive algorithms used appropriate for this overweight and obese cohort?

Lines 149 to 153. Please define the equation abbreviations.

Line 162. Should be Table 2. One decimal place only for energy.

Line 166 Why "XXX"?

Line 196. Parametric testing was used so presumably data were normally distributed. Please state this.

Line 198. The authors have used a two-factor ANOVA (group and time I assume). Why was a repeated measures design not used?

Line 206. There are no data for a control group of no WBC. This is a weakness. Any changes that were observed may have occurred over time irrespective of WBC and differentially in the different BMI groups. This requires comment in the Discussion as a study limitation.

Line 291. Should be Table 3.

Line 298. This picks up the point above. You have no control group on which basis to comment on the effectiveness (or not) of WBC.

Discussion in general. The authors need to address the issue of no control group. E.g. line 338 the authors attribute change to the WBC sessions but changes may have occurred over this duration irrespective of WBC. The authors need to acknowledge this and add a study limitations section.

Author Response

Response to Reviewer 3.

We would like to many thanks for the reviewer for valuable comments that will help improve our manuscript.

The Reviewer wrote: “A generally well written manuscript. Some points to consider.”

Response: Thank you very much for this comment.

The Reviewer wrote: “Line 43. A minor point but "ADP" usually refers to adenosine diphosphate. I wonder if an alternative abbreviation should be used to avoid any possible confusion. Similarly, "WBC" often used for white blood cells.”

Response: Thank you very much for this comment. We know that these abbreviations are ambiguous, but we were guided by the fact that they are used in the available literature (WBC as whole body cryostimulation and ADP as adiponectin). If the reviewer clearly demands changes, we will of course do it.

The Reviewer wrote: “Line 108"Nutritional energy demands".”

Response: Thank you very much for this comment. We changed as requested: “kcal nutritional demands” to “nutritional energy demands”.

The Reviewer wrote: “Line 120. The subject numbers are small. Was a power calculation conducted? If so what sample size was required? 13 out of 40 is a substantial drop out rate. Did the drop outs have any characteristics in common that differed from the remainers? Is this a confounder. Please comment and justify”.

Response: Thank you very much for this comment. The sample size was determined with use of methodology indicated by Suaresh. Data from-prestudy was used. Description was added to the manuscript.

Subjects who resigned from participation in study didn’t have any common characteristics which differed them from the participants who finished the study – we verified it by comparing the available studied parameters in remainders and dropouts.

The Reviewer wrote: “Section 2.3 and Table 1. Although mentioned in the Abstract please state here that the participants were all male.”

Response: We added the data to Table 1 description.

The Reviewer wrote: “Section 2.4. Please provide a reference to the standardized method for circumferential measurements. Were measurements obtained by an ISAK accredited anthropometrist? Weight was measured with the IOI-353 BIA analyser. This provides body composition data yet another analyser, Akern 101, was used to obtain these data - why? Please explain. BIA is a predictive method and is notoriously population specific. Are the predictive algorithms used appropriate for this overweight and obese cohort?”

Response: Thank you very much for this comment. Requested reference was added. Antropometric measurements were performed by an experienced anthropometrist. Each time by the same person, in accordance with ISAK recommendations and book cited in references.

Analysis with Akern BIA device is performed in lying position which allows for stabilization of body fluids level, this type of analysis is more reliable than estimation of body composition in standing position (as in IOI-353 analyzer).

The Reviewer wrote: “Lines 149 to 153. Please define the equation abbreviations.”

Response: Thank you very much for this comment. Abbreviations have been defined as requested.

The Reviewer wrote: “Line 162. Should be Table 2. One decimal place only for energy.”

Response: Thank you very much for this comment. Table 2 has been corrected. Thank you for pointing out this mistake.

The Reviewer wrote: “Line 166 Why "XXX"?

Response: Thank you very much. The name was added instead of XXX.

The Reviewer wrote: “Line 196. Parametric testing was used so presumably data were normally distributed. Please state this.”

Response: Thank you very much for this comment. Statistical analysis description has been corrected.

The Reviewer wrote: Line 198. The authors have used a two-factor ANOVA (group and time I assume). Why was a repeated measures design not used?

Response: Thank you very much for this comment. Repeated measures ANOVA was used, thank you for pointing out this mistake in test name.It is corrected in the text.

The Reviewer wrote: “Line 206. There are no data for a control group of no WBC. This is a weakness. Any changes that were observed may have occurred over time irrespective of WBC and differentially in the different BMI groups. This requires comment in the Discussion as a study limitation”.

Response: Thank you very much for this comment. We have not the results for control group, indeed. It is our study limitation.  We added this in discussion section and in study limitation.

“However, all presented data are associated with differences in response to WBC between people with different BMI. Some changes in tested parameters may be occurs independently of WBC. “

“Our results present only differences in response to WBC people with different BMI. We didn’t investigate changes in tested indicators in people who didn’t use WBC in the same time. “

The Reviewer wrote: “Line 291. Should be Table 3.”

Response: Thank you very much for this comment. Table number has been corrected.

The Reviewer wrote: “Line 298. This picks up the point above. You have no control group on which basis to comment on the effectiveness (or not) of WBC.”

Response. Thank you very much for this comment. We changed this sentence, that …no effect was observed in the same time.

The Reviewer wrote: Discussion in general. The authors need to address the issue of no control group. E.g. line 338 the authors attribute change to the WBC sessions but changes may have occurred over this duration irrespective of WBC. The authors need to acknowledge this and add a study limitations sectionRespose: Thank you very much for this comment. We corrected indicate sentence. In present version is: During experiment no significant changes were observed in BMI…

We added also study limitations: Our study has some limitations. We present only differences between subjects with different BMI during tested period. We didn’t investigate changes in tested indicators in people who didn’t use WBC at the same time. Moreover, the limitation is the size of the studied groups.

Once again, thank you very much for all comments and your time dedicated to us.

Round 2

Reviewer 2 Report

It is now clear when were the blood specimens collected and this is the main problem of this manuscript.

It is not reasonable to compare data obtained at baseline at rest and after cryotherapy sessions that happened one or two weeks later. The authors should have collected blood specimens before and after the first cryostimulation session, as well as before and after the 10th and the 20th session. With such design the authors could have seen how the cryo training may influence the changes in blood analytes due a single cryo session. Moreover, the authors could have seen how this cryo training would influence the concentration of blood analytes at baseline/rest.

In their experimental setting the authors can only make comparison between the data obtained after the 10th and the 20th sessions .

Such a flaw in the design should retain the authors to discuss the impact of a series of cryo sessions.

Author Response

Dear Reviewer,

Thank you very much for your comments. We agree with all comments:

“It is not reasonable to compare data obtained at baseline at rest and after cryotherapy sessions that happened one or two weeks later. The authors should have collected blood specimens before and after the first cryostimulation session, as well as before and after the 10th and the 20th session. With such design the authors could have seen how the cryo training may influence the changes in blood analytes due a single cryo session. Moreover, the authors could have seen how this cryo training would influence the concentration of blood analytes at baseline/rest. In their experimental setting the authors can only make comparison between the data obtained after the 10th and the 20th sessions.”

 Response:

Thus, we made a major revision of our manuscript. We made some corrections in Abstract, Introduction, Results: we add separate figure on which it was presented only rest values and changes remaining figures. We also gave proposal to change tittle of manuscript. We made major revision in Discussion section and Conclusion.

In conclusion, we would like to thank you very much for your in-depth review and valuable manuscript scales.

                                                           With kind regards, MaÅ‚gorzata Å»ychowska

Reviewer 3 Report

The authors have addressed satisfactorily the points raised in my original review.

Author Response

Dear Reviewer, thank you very much for your comments. With Kind Regards, Małgorzata Żychowska

Round 3

Reviewer 2 Report

The quality of the manuscript has improved, and the authors have now compare data which are comparable with each other. However, the design of the study stay perculiar especially with the analyses of blood specimens that were only taken before the first WBC, after the 10th and the 20th WBC.

Some comments

Please write TNF-alpha in full letters throughout the manuscript : the sign « alpha » does not appear correctly.

Abstract :

L30 : °C

What do the authors mean by « extend the WBC procedures. »

Intro :

L77 : Not clear: » however, data regarding the adipokines ADP and LEP are tight ( ??) and elate( ??) to a different number of sessions. »

Figure 1 : « Differences » ?? may be « Blood concentration … »:

L214, L215  and few other one : « reference range »?? or maybe « health related reference interval »

Figure 2 : « normal range »?? or maybe « health related reference interval »

There is a strange extra black square just under (for IOb ,…) concerning the « after 20 ».

L266 : after « Between 10 and 20 WBC » add « there were no»

L294 : levels of

L294 except HDL

L297 : caused a further

L298 were noted

L299 range ?

L312 « Moreover, it is possible that WBC could positively influence on impaired beta-3 adrenergic activity in obesity [16], but this sentence requires further research. » => « Moreover, it is possible that WBC could positively influence on impaired beta-3 adrenergic activity in obesity [16]. »

L314 « the next » ?? please make your message clearer

L321 « Our findings showed a significant decreased in TNF-a between 10 and 20 WBC sessions in 321 all tested groups. » Obviously not with the control group where an increase seems to occur

L362 : TNF alpha

L262 : « All changes in adipokines (especially leptin down-regulation), lipid profile or TNF-alfa may be associated with the changes in activity of the sympathetic system. » =>

… adrenergic receptors, which could cause in lowered metabolic rate in obese subject [58].  Please rephrase

« However, all presented data is associated with the differences in response to 20 compared to 10 WBC between people with different BMI. » Not clear for me, please rephrase

« Some changes in tested parameters may not be dependent of WBC » : what do you mean exactly ?

« Our findings indicate that between 10 and 20 WBC sessions are many positive changes associated with adipokines, lipid profile or body composition. » Not clear

Author Response

Response to Reviewer 2.

Dear Reviewer,

Thank you very much for a thorough check of our article and all your comments.

The Reviewer wrote:

“The quality of the manuscript has improved, and the authors have now compare data which are comparable with each other. However, the design of the study stay perculiar especially with the analyses of blood specimens that were only taken before the first WBC, after the 10th and the 20th WBC.”

Response: Thank you very much. We are aware of the limitations resulting from the number of taken blood samples, it resulted from many reasons, both dependent and independent of us. We cannot improve our manuscript on this issue now, but we will plan and organize the study differently in the future.

Some comments from Reviewer:

The Reviewer wrote:

“Please write TNF-alpha in full letters throughout the manuscript: the sign « alpha » does not appear correctly.”

Response: Thank you very much for this comment. We corrected TNF-alpha according to suggestion in all manuscript.

The Reviewer wrote: „L30 : °C”

Response: Thank you very much. We corrected this mistake.

The Reviewer wrote: “What do the authors mean by « extend the WBC procedures.”

Response: Thank you ery much for this comment.  We changed this sentence. In present version we wrote: Collectively, for people with obesity increasing the number of treatments above the standard 10 should be recommended.

 The Reviewer wrote: “Not clear: » however, data regarding the adipokines ADP and LEP are tight ( ??) and elate( ??) to a different number of sessions. »”.

Response: Thank you very much for this comment. In present version this sentence was corrected. Now is “however, data regarding the adipokines ADP and LEP are few and concern different number of sessions, WBC procedure or BMI of participants”.

The Reviewer wrote: “Figure 1 : « Differences » ?? may be « Blood concentration … »:”

Response: We corrected description of the figure 1 according to suggestion.

The Reviewer wrote: “L214, L215  and few other one : « reference range »?? or maybe « health related reference interval »”

Response: Thank you very much for this suggestion. We changed this according to suggestion.

The Reviewer wrote: “Figure 2 : « normal range »?? or maybe « health related reference interval »“

Response: Thank you very much for this suggestion. We changed this description.

The Reviewer wrote: “There is a strange extra black square just under (for IOb ,…) concerning the « after 20 ».“.

Response: We check all manuscript, and I can’t see this mistake.

The Reviewer wrote: “L266 : after « Between 10 and 20 WBC » add « there were no»”

Response: Thank you very much. We corrected this.

The Reviewer wrote: “L294 : levels of”

Response: We corrected this mistake. Thank you again.

The Reviewer wrote: “L294 except HDL”

Response: Thank you very much. We corrected this.

The Reviewer wrote: „L297 : caused a further”

Response: Thank you, we corrected according to suggestion.

The Reviewer wrote: „L298 were noted”

Response: We corrected according to suggestion.

The Reviewer wrote: „L299 range ?”

Response: Thank you very much. We rewrite it to „health related reference interval”.

The Reviewer wrote: L312: „Moreover, it is possible that WBC could positively influence on impaired beta-3 adrenergic activity in obesity [16], but this sentence requires further research. » => « Moreover, it is possible that WBC could positively influence on impaired beta-3 adrenergic activity in obesity [16].”

Response: Thank you very much. We changed according to suggestion.

The Reviewer wrote: „L314 « the next » ?? please make your message clearer”

Response: We changed this sentence, now is: between 10 and 20 sessions no changes were observed in ADP levels

The Reviewer wrote: L321 « Our findings showed a significant decreased in TNF-a between 10 and 20 WBC sessions in 321 all tested groups. » Obviously not with the control group where an increase seems to occur

Response: Of course, you are right. We change this sentence and now is: Our findings showed a significant decreased in TNF-alpha between 10 and 20 WBC sessions in IOb and IIOb groups.

The Reviewer wrote: „L362 : TNF alpha”

Response: Thank you again. We change TNF-alpha in all manuscript.

The Reviewer wrote: “L262 : « All changes in adipokines (especially leptin down-regulation), lipid profile or TNF-alfa may be associated with the changes in activity of the sympathetic system. » => … adrenergic receptors, which could cause in lowered metabolic rate in obese subject [58].  Please rephrase”

Response: Thank you very much. In present version is: All changes in adipokine (especially leptin) and TNF-alpha levels as well as in lipid profile may be associated with a response of the sympathetic nervous system. It is known that catecholamines, which inhibit production and secretion of TNF-alpha and LEP, are also modulated by sympathetic nervous system. Moreover, animal studies showed that mutations in beta-3 adrenergic receptors lead to lowered metabolic rate in obese subjects [58].

The Reviewer wrote: “« However, all presented data is associated with the differences in response to 20 compared to 10 WBC between people with different BMI. » Not clear for me, please rephrase”

Response: Thank you very much. In present version is: Our data shows a difference in response between 10 and 20 WBCs, however, the effects of greater number of treatments, which may be required for people with class II obesity, were not investigated in this study.

 The Reviewer wrote “« Some changes in tested parameters may not be dependent of WBC » : what do you mean exactly ?”

Response: Thank you very much. In present version of our manuscript, we removed this sentences. All participants had the same condition and procedures of WBC, and balance diet. That was an unfortunate sentence.

The Reviewer wrote : “Our findings indicate that between 10 and 20 WBC sessions are many positive changes associated with adipokines, lipid profile or body composition. » Not clear”

Response: Thank you very much for this comment. In present version we added: Our data show differences due to the use of 10 and 20 WBC however we did not evaluate the greater number of treatments, which may be important for people with II degree of obesity.

Additionaly, we rewrite the first two sentences in method section, concerning blood samples collection: Now is: To assess the initial condition of tested parameters in participants blood samples were collected up to 1 hour before first treatment.  To compare the difference between 10 and 20 treatments blood samples were collected twice: up to 1 hour after 10 and after 20 treatments.

In conclusion, we would like to thank you very much for all valuable comments and your in-depth review and valuable manuscript improvement.

                                                           With kind regards, MaÅ‚gorzata Å»ychowska

Round 4

Reviewer 2 Report

The manuscript has improved.